# Growth dynamics of *Escherichia coli* cells on a surface having AgNbO₃ antimicrobial particles

**Cyrus Talebpour**[ID][1☯]*, **Fereshteh Fani**[2☯], **Hossein Salimnia**[3☯], **Marc Ouellette**[2☯], **Houshang Alamdari**[1☯]

**1** Department of Mining, Metallurgical and Materials Engineering, Université Laval Faculté des Sciences et de Génie, Québec, Quebec, Canada, **2** Department of Microbiology and Immunology, Université Laval Faculté de Médecine, Québec, Quebec, Canada, **3** Department of Pathology, Children's Hospital of Michigan, Detroit, Michigan, United States of America

☯ These authors contributed equally to this work.
* cyrus.talebpour.1@ulaval.ca

## Abstract

The morphological dynamics of microbial cell proliferation on an antimicrobial surface at an early growth stage was studied with *Escherichia coli* on the surface of a gel supplied with AgNbO₃ antimicrobial particles. We demonstrated an inhibitory surface concentration, analogous to minimum inhibitory concentration, beyond which the growth of colonies and formation of biofilm are inhibited. In contrast, at lower concentrations of particles, after a lag time the cells circumvent the antimicrobial activity of the particles and grow with a rate similar to the case in the absence of particles. The lag time depends on the surface concentration of the particles and amounts to 2 h at a concentration of ½ minimum inhibitory concentration. The applicability of these findings, in terms of estimating inhibitory surface concentration, was tested in the case of antimicrobial polymethyl methacrylate (PMMA) bone cement.

**Data Availability Statement:** All relevant data are within the manuscript and its Supporting information files.

## 1. Introduction

The misuse of antibiotics has led to development of infections attributed to antimicrobial resistant (AMR) bacteria strains, which have been responsible for an annual 1.3 million deaths globally [1]. If nothing is done, this figure is projected to rise to 10 million by 2050 [2], with expenditures resulting in a global loss of $108 trillion USD [3]. The need to counter AMR may be achieved through strategies such as effectively reducing the microbial load on surfaces that come into contact with pathogenic microbial cells [4]. Any remedy for rendering antimicrobial properties to these surfaces is desired to present no systemic or localized toxicity, inhibit the proliferation of most species of microbial cells in contrast to a selected category of species, provide a robust system enabling ease-of-use, and involve a low production cost [5]. In this respect, silver coatings have been good candidates [6–8], thanks to their broad-spectrum antimicrobial activity through mechanisms such as production of reactive oxygen species (ROS) [9]. However, this approach is associated with drawbacks such as incompatibility of the conventional coating techniques with some important solids, altering the structure of surface, and corrosion with elution of silver ions, which ironically are the agent of antimicrobial activity [10].

**Funding:** This study was financed by a Natural Sciences and Engineering Research Council of Canada (https://www.nserc-crsng.gc.ca/index_eng.asp) grant No. RGPIN-2023-04795 to H.A. and by the Canadian Institutes of Health Research (https://cihr-irsc.gc.ca/e/193.html) foundation grant No. FND167283 to M.O. M.O. is the holder of a Canada Research Chair in Antimicrobial Resistance. The sponsors did not play any role in the study design, data collection and analysis, decision to publish, or preparation of the manuscript.

**Competing interests:** The authors have declared that no competing interests exist.

Recently, we found a possible solution for the latter drawback by synthesizing $AgNbO_3$ particles, which, despite having diminished ion release rate, show strong antimicrobial activity [11]. This kind of antimicrobial activity, which we had defined as 'through contact', are associated with particulate agents and can be incorporated into the surface provided that the required amount of the particles is not below a threshold. Thus, a main question to be addressed for designing an antimicrobial surface employing these particles is introducing and quantifying a property akin to the minimum inhibitory concentration (MIC).

Another question, relevant for the antimicrobial surfaces in niche applications such as the interfacial surfaces of implants, is the formation and evolution of biofilms, i.e., structured communities of sessile bacteria [12] encased in a self-produced matrix of extracellular polymeric substances (EPS) [13]. In the context of AMR, the biofilm structure facilitates quorum sensing between cells responsible for altering gene expression, accumulating genetic material for cellular uptake, promoting horizontal gene transfer, and enabling less metabolically active cells known as persister cells to arise [14]. Up to 65–80% of all infections are associated with biofilm formation, much of which are implicated in chronic infections, in contrast to planktonic bacteria involved in acute processes [15]. Therefore, a phenomenological investigation of biofilm evolution on a surface having a concentration of antimicrobial particles sublethal to bacteria is a minimum necessity for actual implementation of an antimicrobial surface.

In this study we determine the surface minimum inhibitory concentration (SMIC) for surface-utilizing particles with antimicrobial activity through contact. The utility of these findings is illustrated in the case of polymethyl methacrylate (PMMA) bone cement. Furthermore, a qualitative description of colony formation and expansion at early stages are presented by time lapse microscopy on a model system, consisting of a gel having dispersed $AgNbO_3$ particles on its surface. It is worth mentioning that incorporation of the right amount of $AgNbO_3$ nanoparticles in PMMA bone cement is expected to confer permanent antibacterial property to the cement due to its extremely limited corrosion/dissolution rate, contrary to the incorporation of conventional antibiotics such as gentamicin, as verified in our recent publication [16]. There, we demonstrated the persistence of antibacterial properties for a PMMA sample mixed with 1% $AgNbO_3$ particles (w/w) over a four-month period with no reduction in its potency when kept in water.

## 2. Methodology

### 2.1 Preparation and characterization of antimicrobial particles

$AgNbO_3$ antimicrobial particles were synthesized using the Activated Reactive Synthesis (ARS) method as described previously [11, 17]. Briefly, the raw materials $Ag_2O$ (Sigma-Aldrich Corp) and $Nb_2O_5$ (Inframat® Advanced Materials LLC), with the weight ratio of 1 g to 1.147 g respectively (for each 1 g of $Ag_2O$, 1.147 g of $Nb_2O_5$ powders), were mixed in a hardened steel crucible with high energy ball milling for 10 min. The mixture was transferred to a ceramic crucible and placed in an oven where it was gradually heated at a rate of 5˚C/min until the formation temperature, 1000˚C, was reached. The mixture was kept at this temperature for about 4 h and gradually cooled down at a rate of 10˚C/min to room temperature. The post-synthesis treatment included subjecting the powder to successive steps of high and low energy ball milling. The high energy ball milling was carried out using the 8000D Mixer/Mill® (SPEX SamplePrep, LLC). An 8001 hardened steel crucible and a milling media consisting of two balls of 12.70 mm and one ball of 6.35 mm in diameter were used for milling 7 g of material in each batch at an oscillation rate of 1060 cycles/min for a duration of 90 min. Low energy ball milling was carried out by transferring approximately 40 g of powder from the previous step (agglomerates) to a crucible containing hundreds of steel beads of 4.5 mm in diameter, which were

made to rotate at 90 rpm by Szegvari Attritor System Type E Model 01-STD (Union Process, Inc.). To this, 10 mL of water was added and the attrition process was performed for 120 min. At the end of the operation the beads were rinsed with deionized water and the residue thus obtained was dried inside an oven with a temperature of 150˚C overnight to obtain the final nanostructured $AgNbO_3$ particles. The particles were previously characterized in terms of size by the Malvern ZEN1600 Dynamic Light Scattering (DLS) machine and found to have an average size of 0.44 μm [11]. Density of the particles was measured using a AccuPyc II 1340 (Micromeritics Instruments Corp.) helium gas pycnometer. Powder samples of ~ 8 g were weighted in a 10 mL stainless steel cell, and the density was obtained by dividing the mass of the sample to the volume measured by the pycnometer.

## 2.2 Bacterial strain and culture conditions

The bacterial strain used in this study was *Escherichia coli* ATCC # 25922 and was prepared as described below. Cell stock in 50% glycerol was taken from a -80˚C freezer and after thawing, 30–50 μL of it was transferred to a Tryptic Soy Agar (TSA) (with 5% sheep blood) plate (the P1 plate), inoculated by standard streaking, and incubated at 37˚C overnight under aerobic condition until colonies were visible. A well-formed representative colony from the plate was picked and inoculated into 3 mL of Tryptic Soy Broth (TSB) and incubated at 37˚C with 150 rpm shaking for 3 h. Then, a 1 mL culture was transferred to 2 mL autoclaved tubes and centrifuged at 8000 rpm for 8 min in a microcentrifuge to wash out particulate debris, which may appear as artifacts during imaging. The harvested cells were resuspended in 1 mL TSB and 0.5 mL was used for $OD_{600}$ measurement. Using an in-house $OD_{600}$ vs cell count correlation database, a cell suspension of $10^8$ CFU/mL was prepared. The suspension was then serially diluted in TSB to a final concentration of $10^7$, $10^6$ and $10^5$ CFU/mL by pipetting and vortex mixing at each dilution step.

## 2.3 Planktonic growth curve and antimicrobial susceptibility

The growth of *Escherichia coli* ATCC # 25922 bacterial cells were determined using an automated incubation system (Biospa, BioTek) integrated with Cytation 5 multimode reader (BioTek). 10 μl of $1 \times 10^7$ CFU/mL bacteria cells were inoculated in 990 μl of LB medium in the absence or presence of $AgNbO_3$ particles at concentrations of 4, 8, 16 and 32 μg/mL in a Falcon® 24-well plate. The plate was incubated at 37˚C, and the $OD_{600}$ was read at each 1 h for 20 h after shaking for 10 s using a Cytation 5 multimode reader. Data was processed using Gen5 software and reported with GraphPad Prism. The minimum inhibitory concentration (MIC) was determined as the concentration of particles for which no growth was observed after 24 h of incubation. This approach was adopted from the methodology presented by Haque et al. [18]. All MICs were determined from at least three independent biological replicates.

## 2.4 Preparation of antimicrobial gel surfaces

5 mg of the particles was weighted and resuspended in 5 mL of deionized water. The suspension was serially diluted to concentrations of 200, 100, and 50 μg/mL. To prepare an antimicrobial gel, standard A9 blood agar plate was left on the table for 30 min, then 5 μL of each particle suspension, after vortexing for 1 min, was dispensed at a spot in the first half of the plate. The procedure was repeated on the second half of the plate to obtain two technical replicates. On each half of the plate, 5 μL of water was dispensed on designated spots as control. The dispensed particle suspension spreads to a disk with a diameter of ~ 8 mm and surface area of ~ 50 mm$^2$ before drying after 30 min. Thus, the particle density at the respective spots corresponding to 50 μg/mL, 100 μg/mL, and 200 μg/mL particle suspensions are respectively 5 ng/mm$^2$, 10 ng/mm$^2$, and 20 ng/mm$^2$. Three biological replicates were prepared for each condition.

## 2.5 Plating microbial cells and interrogating the morphological evolution and size of the resulting microbial cell colonies

From a $1 \times 10^5$ CFU/mL *Escherichia coli* ATCC # 25922 cell stock, 1 μL (nominally containing 100 CFU) was dispensed on each spot wherein particle suspensions and controls were previously dispensed. The dispensed cell stock spreads to a disk with a diameter of ~ 4 mm and surface area of ~ 12 mm$^2$. The plates were placed inside the incubator at 37°C for 6 h. Meanwhile, a rectangular part at the center of the spots of dispensed bacteria, having an area of 2.05 mm$^2$ (1.75 mm × 1.17 mm), were imaged with a metallurgical microscope (10× objective) in a bright and dark field at different time intervals. Each image represents approximately 1/6 of the entire spot area and the number of the growing colonies detected on it fairly well approximates 1/6 of the dispensed cells, verifying the homogenous dispersion of the bacteria cell on the surface. The area of 20 colonies, from the images of the two spots for each condition, was selected using ImageJ software. Each condition represents a given combination of antimicrobial particle concentration and incubation time. Then, the average area was calculated for determining the average colony size for that given condition. For each condition, three representative colonies were selected for extracting characteristic features.

## 2.6 Preparing and testing antimicrobial disks

AgNbO$_3$ loaded polymethylmethacrylate (PMMA) disks were prepared by mixing 10 g of the PMMA dry component (A formulation of 97% Poly(methyl methacrylate), Sigma-Aldrich, and 3% Benzoyl peroxide, Sigma-Aldrich) with different amounts of AgNbO$_3$ (Control, 0.5%, 1%, and 2%) to obtain the desired W/W ratio. Then, 5 mL of liquid monomer (A formulation of 97% Methyl methacrylate, Sigma-Aldrich, and 3% N,N-Dimethyl-p-toluidine, thermo scientific) was added to the mixture and mixed until a homogenous paste was formed. This was immediately cast in a High-Density Polyethylene (HDPE) mold with an inner diameter of 12 mm and depth of 10 mm. After solidification, the disc was released from the mold and used in subsequent experiments. The naming conventions for the samples are hereafter designated as control PMMA and PMMA—X% AgNbO$_3$ (X = 0.5, 1, 2).

The disks were tested for antibacterial properties. Briefly, three replicates of each disk were disinfected by soaking in 70% isopropanol for one min, washing in sterile water for 2 min, and allowing them to air dry. The top surfaces of the disks were respectively loaded with a 50 μl of $1.5 \times 10^6$ CFU/mL *Escherichia coli* ATCC # 25922 bacterial cell suspensions (corresponding to around 75000 bacterial cells) and, after allowing to dry in air for 3 h, were dropped in a tube of 5 mL TSB. 5 min after dropping the disks in TSB, a 1 mL sample from each growth media was transferred in a Falcon ⓡ 24-well plate. The original tubes were incubated at 37°C overnight. After incubation, the tube was checked visually for signs of growth (turbidity) or lack of growth (clear medium). For the plate, the OD$_{600}$ was read each 1 h for 20 h after shaking for 10 s using a Cytation 5 multimode reader (BioTek).

## 3. Results and discussion

### 3.1 Planktonic growth dynamics

Fig 1 presents the planktonic growth dynamics, indicated by average optical density over three biological replicates versus incubation time. Based on the general shape of the curves, the impact of antimicrobial particles on the cell viability, being manifested in the form of prolonging the lag phase by 4 and 8 h, respectively for 4 and 8 μg/mL samples. This may be a manifestation of the "tolerance by lag" phenomenon, which appears to involve a generalized adaptive response to antibiotic stress [19]. Alternatively, one may attribute the prolonged lag phase to

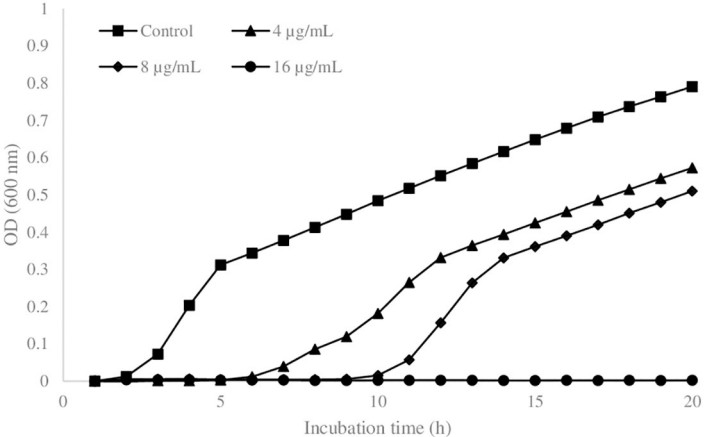

**Fig 1. The growth curve of *Escherichia coli* ATCC # 25922 in LB broth in the presence of antimicrobial AgNbO₃ particles with different concentrations.** Each point is the average of three biological replicas.

the presence of a subpopulation of antimicrobial persister cells in the entire inoculum $10^5$ CFU that have a phenotypic tolerance to the antimicrobial particles [20]. The second key observation is that the minimum inhibitory value is at most 16 μg/mL, as there is no sign of growth (increase in $OD_{600}$) for this level of particle concentration after 20 h of incubation. The same lack of increase in $OD_{600}$ was seen for the case of AgNbO₃ concentration of 32 μg/mL, which was not included in Fig 1 due to overcrowding. Regarding these observations, some contrasts are expected with the case of exposing the microbial cells to antimicrobial particles on the surfaces for the following reasons. As the action of these particles is exerted via contact (since they do not release silver ions over the MIC level [11]) the growth inhibition on the surface has to occur either by incidental placing of a seeded cell at the proximity of a particle or via the cell motility leading to collisions.

## 3.2 Morphological and growth dynamics of colonies on a surface with no antimicrobial particles

The growth behavior of colonies on a plate with no antimicrobial particles performed as an independent test was selected as a baseline for assessing the impact of particles during the following experiments. The results are hereafter referred to as "trend". The parameters selected for this purpose were colony shape and its overall size, which is defined as the diameter, *d*, of a disk with similar area, as specified in Eq 1.

$$d = 2\sqrt{\frac{Area}{\pi}} \tag{1}$$

When 1 μL of a $1 \times 10^2$ CFU/μL cell suspension was dispensed on the agar plate, 22 and 24 colonies were observed on the respective images taken from two spots. The area of the images of the two spots represented $1/6^{th}$ of the total seeded area. The average of these two numbers, i.e., 23, indicates the number of colonies in the examined area. The total number of the colonies can then be estimated as $6 \times 23 = 138$, which is close to the nominally expected number of the dispensed cells. Fig 2 presents zoomed images of a typical selected colony taken at different times of incubation of (2–5) h. The contours of the colonies at different time intervals are superimposed and presented at the center of the figure. The images may be used to describe

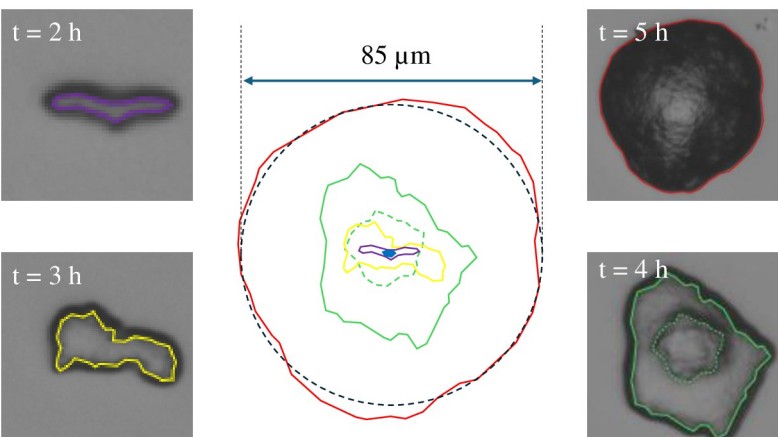

**Fig 2. The morphological dynamics of three colonies grown on the surface of an agar gel and monitored with time lapse microscopy over 5 h of incubation.** The contours indicate the colony boundary in the following order: Blue oval in center: Colony at t = 1 h, Violet: 2 h, Yellow: 3 h, Green: 4 h, and Red: 5 h.

the phases of colony morphology. In the first 3 h of incubation, the colony has expanded into an irregular shape, which, based on dark field images (not presented here), is in the form of a monolayer sheet of cells. At t = 3 h, the area of the colonies, averaged over the 10 colonies across the images of the two spots, is 391 $\mu m^2$. After an extra hour of incubation, the average colony area increases to 2044 $\mu m^2$, which is an increase by a factor of 5.2. By taking the log base 2 of 5.2, we get ~ 2.4 division cycles. This factor is significantly smaller than the growth rate of 2.8 determined for microbial cells within a microcolony on an agar plate (See S1 Appendix). The deviation can be explained by noting that a second layer with a smaller area (presented by dashed green curve on the image corresponding to t = 4 h) was formed at the inner part of the first layer, thus increasing the total biomass of the colony by ~ 0.4 cycles. Further incubation time increases these inner layers and the incremental change in colony's apparent area drops to 3.6× (~ 1.9 cycles). This indicates a transition from the early stage of single layer colony to the stage when the colony is in the form of an incomplete cone with many layers [21, 22]. Based on the published research in the literature, the described trend in colony morphology results from the bacterial motility [23–27] and formation of extracellular matrix and/or cellulose which might contain protein fibers [28].

This qualitative observation suggests a phenomenological description of a colony as schematically presented in Fig 3. The colony starts with a rugged monolayer cell as suggested by a single curved bright boundary in the accompanying dark field image. Then, distinct cell layers are added on the top of the first layer (z-direction). As the colony grows further, more layers are added and their boundaries in the dark-field image lose distinction. According to this model, the growth on solid phase takes place in 3 stages: 1) Early development phase, during which the colony is in the form of a single layer cell sheet and does not yet exhibit clear characteristic features, 2) Intermediate development phase, during which distinct cell layers are formed over the first layer, and, 3) Mature phase, when the layered structure makes a transition to a rough mass of cells. Based on this view, during the early development phase the cells are in direct contact with the surface and if the antimicrobial activity of the surface is not sufficient to prevent cell proliferation in a time scale of less than a few growth cycles, the upper layers may form. If the antimicrobial activity is exerted by contact, one can anticipate that these top layers be protected from the antimicrobial surface via the first layer, which acts as a barrier.

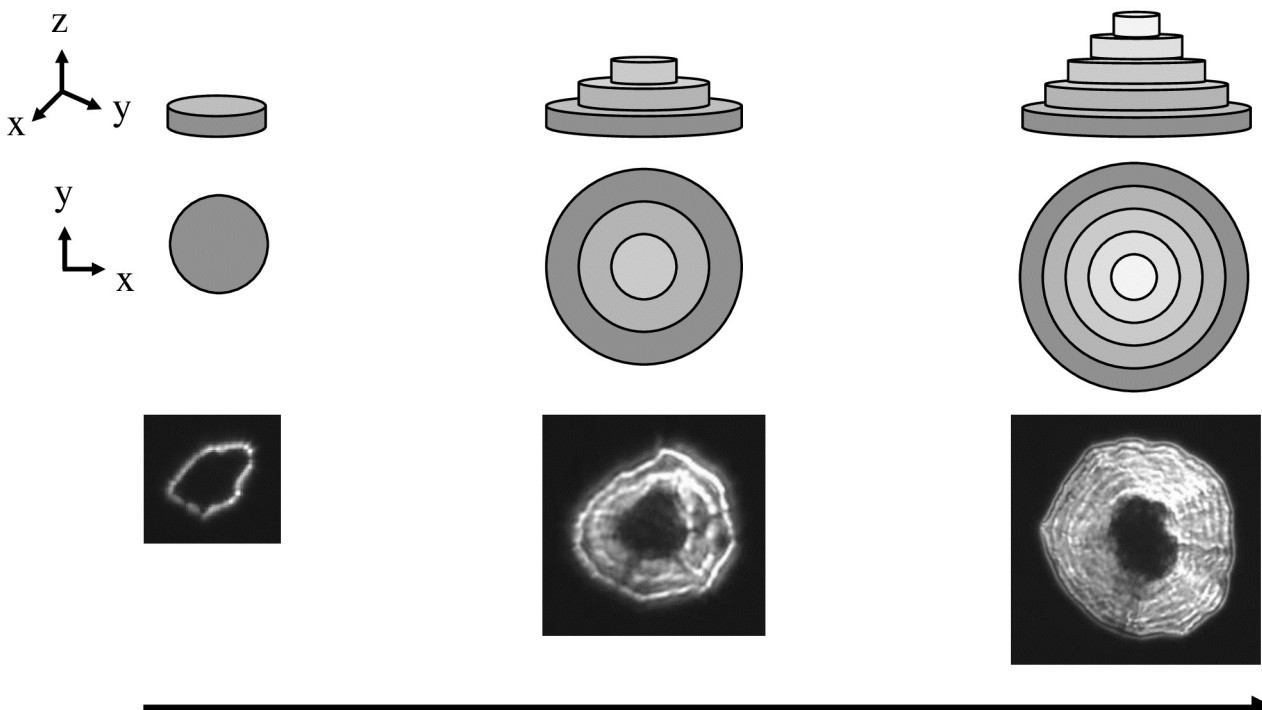

**Fig 3. The schematic layered structure of a typical colony over incubation time.** Schematics adapted from the reference [29], along with recorded dark-field images of a typical colony.

### 3.3 Growth dynamics of colonies on a surface containing antimicrobial particles

Fig A-D in S6 Appendix presents the partial image of the inoculated spot for different surface concentrations of antimicrobial particles of (0, 5, 10 and 20) ng/mm$^2$ cases. The presented images are of after 5 h of incubation specifically because at this time point the colonies are comparatively large but their characteristic morphology has not been perturbed due to formation of multiple layers and extracellular matrix. In addition, at the next point (6 h of incubation) some colonies merge with their neighbors.

The images of the spots after leaving the plates at room temperature overnight, are presented in Fig 4. The total number of colonies and the sizes of colonies grown on the two images for each condition after 6 h incubation were averaged and presented in Table 1. As it is seen the difference in colony sizes are statistically significant.

The following observations were made: 1) 15% of the dispensed cells did not proliferate on the gel having 5 ng/mm$^2$ of AgNbO$_3$ particles. 2) The viability of the dispensed cells on the gel having 10 ng/mm$^2$ of particles was 19%, and 3) There is no sign of growth for 20 ng/mm$^2$. Accordingly, we took 20 ng/mm$^2$ as equivalent to the MIC in the case of planktonic growth represented by the curves in Fig 1, albeit this does not imply an equivalency as it is the case for microdilution and agar dilution susceptibility tests for antibiotics [30]. The discrepancy results from the fact that the antimicrobial particles interact with bacterial cells not by diffusion but via contact.

Assuming an actual size of 0.69 μm for particles (See S3 Appendix), the surface MIC of 20 ng/mm$^2$ translates to ~ 2.7 mg/mL of AgNbO$_3$ (See S4 Appendix); a value much higher than

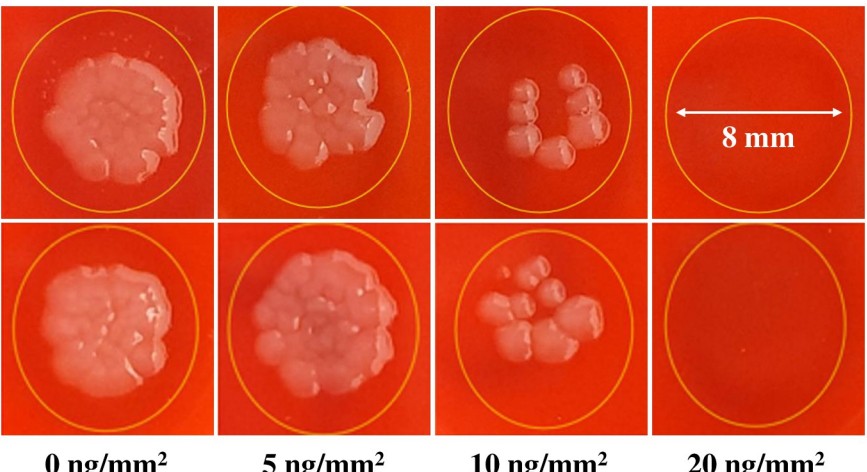

**Fig 4. Shape of *Escherichia coli* colonies in two separate inocula on different BAP plates after leaving at room temperature overnight with varying concentrations of AgNbO$_3$.**

the 16 µg/mL obtained for planktonic cells (Fig 1). As it will be later seen, this observation is in good agreement with the exemplary case of loading a composite solid with the antimicrobial particles. However, we first report our observation with the growth behavior of bacteria on the surface of gels containing sub-MIC surface concentration of antimicrobial particles.

In Fig 5, the size of microcolonies (average and standard deviation over 10 representative colonies) are presented as a function of time and were compared to those measured for the trend obtained in an independent test, as described in the previous subsection. Three observations can be made from this figure: 1) As it is expected, the growth behavior of the bacteria on a gel, as indicated by average colony size, is well repeated (control vs trend), 2) the lag time increases with concentration of antimicrobial particles. However, the lag time is smaller on surfaces than that in planktonic cases. For example, at ¼ and ½ the surface MIC, i.e. gels with 5 ng/mm$^2$ and 10 ng/mm$^2$ nanoparticles, the lag phases are 1 and 2 h respectively (Fig 5). In the planktonic cases, at antimicrobial concentrations of ¼ and ½ the MIC, the lag phases are 4 and 8 h respectively. 3) Finally, and similarly to planktonic cells (Fig 1), the cells whose growth

**Table 1. The statistics on the sizes of colonies grown on two 1.75 mm × 1.17 mm rectangular sections of two inoculated spots, having different surface density of particles, after 6 h incubation.**

| Surface concentration | 0 ng/mm$^2$ | | 5 ng/mm$^2$ | | 10 ng/mm$^2$ | | 20 ng/mm$^2$ | |
|---|---|---|---|---|---|---|---|---|
| Number of colonies (Per image) | Img. 1 | Img. 2 | Img. 1 | Img. 2 | Img. 1 | Img. 2 | Img. 1 | Img. 2 |
| | 22 | 19 | 18 | 17 | 4 | 3 | 0 | 0 |
| Number of colonies (Total) | 41 | | 35 | | 7 | | 0 | |
| Average size (µm) | 150 | | 77 | | 48 | | Not determined | |
| Standard deviation | 16 | | 19 | | 14 | | Not determined | |
| Two-sample T-test for 0 ng/mm$^2$ and 5 ng/mm$^2$ | | | | | | | | |
| T-value | | | Degree of freedom | | | | p-value | |
| 18.18 | | | 74 | | | | 0 | |
| Two-sample T-test for 5 ng/mm$^2$ and 10 ng/mm$^2$ | | | | | | | | |
| T-value | | | Degree of freedom | | | | p-value | |
| 3.82 | | | 40 | | | | 0.0004558 | |

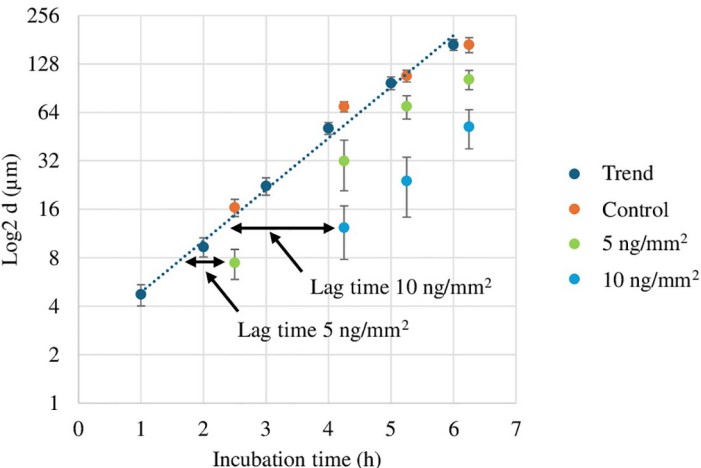

**Fig 5. The average size of microbial cells grown on a gel surface having antimicrobial particles with surface densities of 5 and 10 ng/mm² as compared to the size of colonies grown on a normal gel.**

was not inhibited, after a lag time, proliferated with an exponential growth rate, similar to the case of no antimicrobial particles (Fig 5).

In Fig 6, the cropped microscopic images and time-lapse contours at 2 h, 4 h, 5 h, and 6 h of three colonies for control, 5 ng/mm² and 10 ng/mm² gels are presented. A dashed circle representing a radius of 150 μm, 100 μm and 70 μm was overlaid on the time-lapse contours of colonies from the control, 5 ng/mm² and 10 ng/mm² of $AgNbO_3$ respectively. Within each of these dashed circles, the location of antimicrobial particles are presented as solid gray points. The original microscopic image of the gels after 5 h of incubation showing the location of the colonies are presented in Fig A-C in S6 Appendix.

In the case of the control gel, similar to the phenomenological description of colonies in the independent baseline test, the colony starts with a cell monolayer and its size logarithmically increases. After 4 h of incubation, more cell layers are added over the first layer and finally the colony matures by the formation of a rough biomass structure. For the case of growth on gels with antimicrobial particles, the general trend is similar to the control, except the events have been delayed by lag time. This implies that once a cell overcomes the antimicrobial induced lag phase, it is able to proceed with proliferation as part of a biofilm. To elucidate this point, in Fig 7 we have presented the photo of a colony (the second colony of the second row in Fig 6) after 5 h of incubation. We have overlaid the photo with circles representing particle aggregates and counters of it at earlier times of 2 and 4 h, the preparation of which has been described with details in S7 Appendix.

In the photo, we have marked 4 points A1 to A4 located at the contour of the Colony at time t = 4. Based on intuitive knowledge from the qualitatively analogous dynamics of a spreading droplet on a smooth surface [31], we assumed that these borderline points would be respectively shifted to points B1 to B4 along paths indicated by displacement vectors, whose outward directions are perpendicular to the contour. Interestingly, the magnitudes of these vectors are approximately similar despite the fact that there were particle aggregates along vectors A1B1, A2B2 and A3B3, while the A4B4 was free of aggregates. A plausible explanation for this observation could be the fact that the exposure time of the cells at outwardly expanding areas to antimicrobial particles is too short to impact the proliferation. Concerning drug-bacteria interaction dynamics, Frenkel et al. describe the bistable effect, i.e.

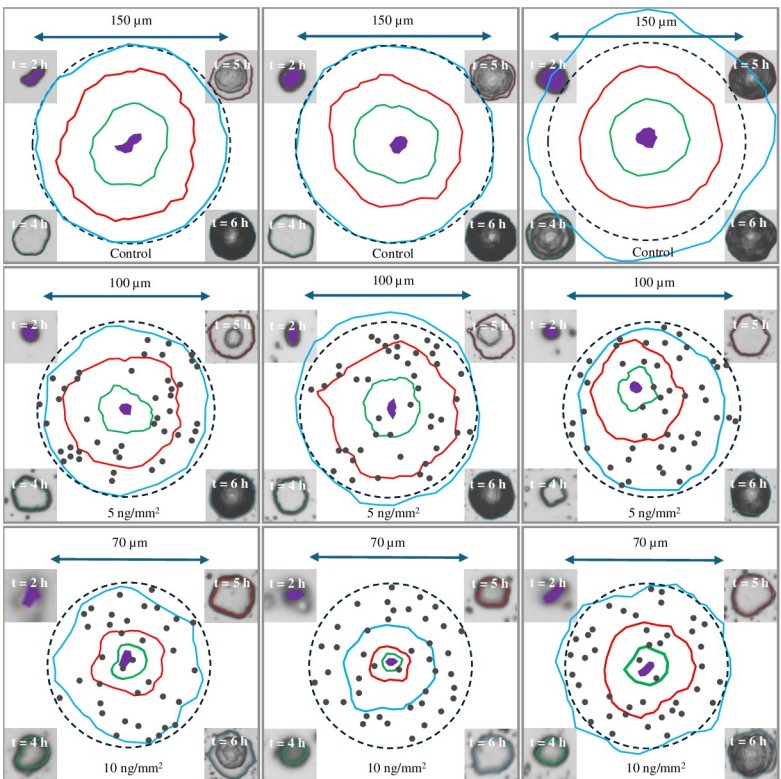

**Fig 6. Evolution of the morphology of three representative colonies in the presence of different antimicrobial particle concentrations.** First row: no antimicrobial particles, Second row: having antimicrobial particles with a surface density of 5 ng/mm$^2$, and Third row: having antimicrobial particles with a surface density of 10 ng/mm$^2$. Color code: Violet (2 h), Green (4 h), Red (5 h), and Cyan (6 h). The images of the colony after different incubation times are also shown at the corners in each case (starting from 2 h at upper left corner moving counterclockwise: 2 h, 4 h, 5 h, and 6 h). The antimicrobial particles are represented by solid gray points.

a system which admits more than one stable state [32]. For example, given a sub-MIC level of antimicrobial agent, there remains still the possibility for the bacteria to eventually proliferate, depending on factors such as nutrient availability of the medium, the potential degradation of the antimicrobial agent by the bacteria, or the genetic/phenotypic heterogeneity of the bacteria population.

According to our observations, we hypothesize that just after being inoculated, the cell starts preparing for the exponential growth phase by trying to accumulate transient metals [33]. The presence of a silver-containing compound in the vicinity perturbs this process. However, if a cell could overcome this hurdle and successfully divide, its descendants produce EPS, which acts as a structural barrier and provides tolerance to the antimicrobial particles [34]. In other words, the surface density of the particles must be sufficient enough to prevent the proliferation from the beginning, otherwise bacteria will create a protective barrier and continue to grow. This finding is of special importance in designing appropriate antibacterial surfaces, namely the bone cements which is our main goal. The bone cement must have sufficient particle density at its surface to prevent the proliferation of bacteria cells but not too high to be safe for human cells.

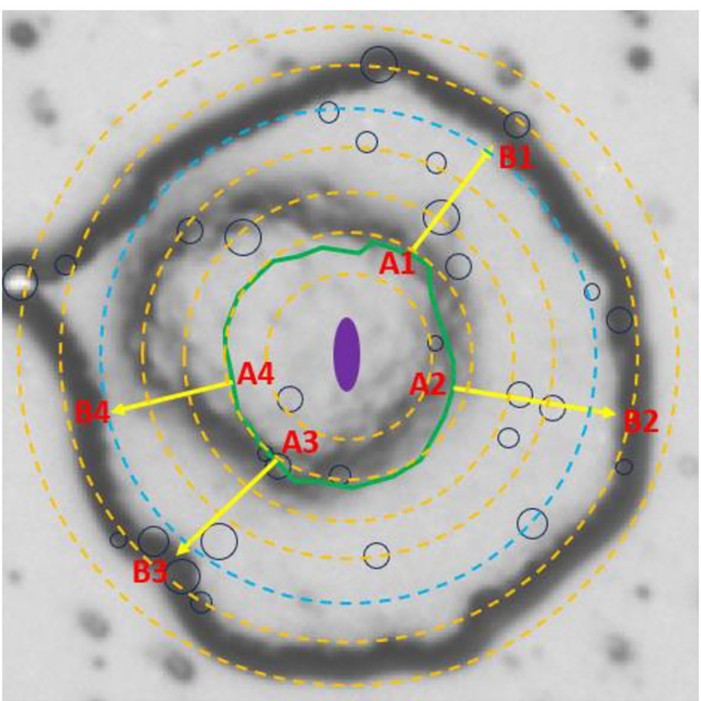

**Fig 7. The photo of a typical colony after 5 h of incubation on a spot with particle concentration of 5 ng/mm².** The photo is overlaid with particle aggregates represented by proportionally sized circles. Also represented is the contour of the colony at earlier incubation times (violet 2 h, green 4 h). The dashed scale circles start from d = 10 μm for the innermost and continue and continue with 10 μm increment ending with d = 70 μm for the outermost. The diameter of the dashed blue circle is 50 μm.

## 3.4 Bacterial proliferation on the surface of a composite solid loaded with antimicrobial particles

Our findings on the gel surface MIC was used as a an insight for estimating the antimicrobial level of a composite material. In this regard, we used PMMA as a solid, which is widely used as bone cement in orthopedic operations. The PMMA disks, loaded by different amounts of AgNbO$_3$ particles, were inoculated with ~ 75000 CFU of *Escherichia coli* and the survival of the cells were assessed. The result is summarized in Fig 8. The first observation is the 3 h of lag in the case of the PMMA—0.5% AgNbO$_3$ bone cement. This is more similar to the ~ 2 h lag for the 10 ng/mm² gel (as indicated in Fig 5) than to the ~ 8 h of lag near MIC during the planktonic growth case of Fig 1. The second key observation is related to the surface MIC at the PMMA—1% AgNbO$_3$. According to the data presented in S2 Appendix, the estimated surface density of the AgNbO$_3$ particles for 20 ng/mm² gel is $1.93 \times 10^4$ particles/mm². If the particles are loaded in the bulk gel, this would correspond to an equivalent volume of $2.68 \times 10^{-3}$ g/mL loading (See S4 Appendix). The mass density of PMMA is 1.18 g/mL [35]. If the bone cement surface behaved exactly similar to the particle loaded gels, $2.68 \times 10^{-3}$ g/mL loading would correspond to $(2.68 \times 10^{-3}$ g/1.18g$) \times 100\% = 0.23\%$ of the cement, in w/w. The value is indeed not far from the observed 1% w/w for the particle loaded PMMA when the impact of inhomogeneous particle distribution on the surface is taken into consideration. More details on this respect are provided in S5 Appendix.

Another factor, which contributes to the minor discordance between the antimicrobial efficacy of the antimicrobial particles on gel surface and those loaded into the bone cement, is the

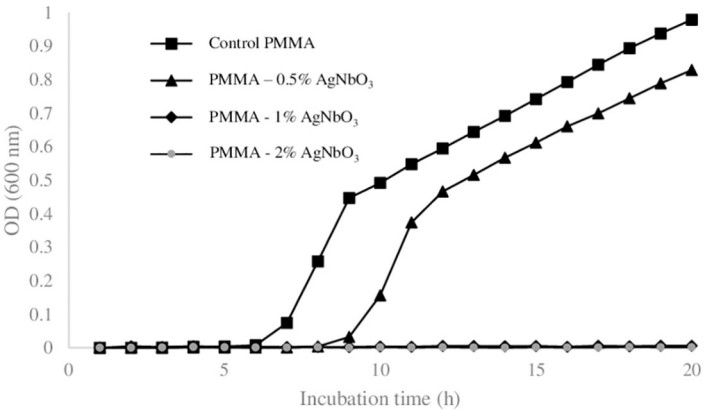

**Fig 8. The optical density of the broth in which the *Escherichia coli* ATCC # 25922 cells contacted with the surfaces of PMMA samples, having different amounts of AgNbO₃, are incubated.** Each data point is average over 3 biological replicas.

dissimilarity of inoculum in the two cases. We used fewer cells on the gel to avoid crowding which causes the merging of the colonies. On the other hand, we used a higher amount of cells on the cement surface to increase our confidence in calling the surface a true antimicrobial. Nonetheless we believe that the experiments on the gel are a good proxy for the microbial proliferation on the antimicrobial surfaces produced by loading antimicrobial particles.

## Conclusion

In conclusion, the growth of microbial colonies on the surface of a gel loaded with antimicrobial AgNbO₃ particles was investigated. It was found that similar to the planktonic case, the impact of sublethal levels of the antimicrobial particles is manifested as prolonged lag phase and a corresponding surface minimum inhibitory concentration (SMIC) could be defined above which the bacterial cells do not form colonies. The SMIC obtained in this way is a good estimator for the required antimicrobial concentration to be loaded into a solid to provide its surface with the antimicrobial property.

## Supporting information

**S1 Appendix. Measuring solid phase growth rate.**
(DOCX)

**S2 Appendix. Surface density of AgNbO₃ particles on antimicrobial gels.**
(DOCX)

**S3 Appendix. Estimating average particle size, volume and mass.**
(DOCX)

**S4 Appendix. Determining the equivalent MIC.**
(DOCX)

**S5 Appendix. Determining distance between particles.**
(DOCX)

**S6 Appendix. Supplementary information for Fig 6.**
(DOCX)

**S7 Appendix. More data on the evolution of colonies.**
(DOCX)

**S8 Appendix. Size distribution and density of the antimicrobial particles on the bone cement.**
(DOCX)

## Author Contributions

**Conceptualization:** Cyrus Talebpour.

**Formal analysis:** Cyrus Talebpour, Houshang Alamdari.

**Funding acquisition:** Marc Ouellette, Houshang Alamdari.

**Investigation:** Cyrus Talebpour, Fereshteh Fani, Hossein Salimnia.

**Methodology:** Cyrus Talebpour, Fereshteh Fani, Hossein Salimnia.

**Project administration:** Marc Ouellette, Houshang Alamdari.

**Resources:** Hossein Salimnia, Marc Ouellette, Houshang Alamdari.

**Software:** Cyrus Talebpour.

**Supervision:** Houshang Alamdari.

**Validation:** Fereshteh Fani, Hossein Salimnia.

**Visualization:** Cyrus Talebpour.

**Writing – original draft:** Cyrus Talebpour.

**Writing – review & editing:** Cyrus Talebpour, Fereshteh Fani, Hossein Salimnia, Marc Ouellette, Houshang Alamdari.

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
