## [Decision Letter · Decision Letter 0]

24 Jun 2024

PONE-D-24-21073Bacterial growth dynamics on a surface having a particulate antimicrobial agentPLOS ONE

Dear Dr. Talebpour,

Thank you for submitting your manuscript to PLOS ONE. After careful consideration, we feel that it has merit but does not fully meet PLOS ONE’s publication criteria as it currently stands. Therefore, we invite you to submit a revised version of the manuscript that addresses the points raised during the review process.

We look forward to receiving your revised manuscript.

Kind regards,

Hae Lin Jang

Academic Editor

PLOS ONE

Journal Requirements:

2. Please amend either the abstract on the online submission form (via Edit Submission) or the abstract in the manuscript so that they are identical.

Reviewers' comments:

Reviewer's Responses to Questions

**Comments to the Author**

1. Is the manuscript technically sound, and do the data support the conclusions?

Reviewer #1: Yes

Reviewer #2: Yes

Reviewer #3: Partly

2. Has the statistical analysis been performed appropriately and rigorously? 

Reviewer #1: N/A

Reviewer #2: Yes

Reviewer #3: Yes

3. Have the authors made all data underlying the findings in their manuscript fully available?

Reviewer #1: Yes

Reviewer #2: Yes

Reviewer #3: Yes

4. Is the manuscript presented in an intelligible fashion and written in standard English?

Reviewer #1: Yes

Reviewer #2: Yes

Reviewer #3: Yes

5. Review Comments to the Author

Reviewer #1: The manuscript entitled "Bacterial growth dynamics on a surface having a particulate antimicrobial agent' was evaluated. I suggest the manuscript for publishing, before it can be accepted it should revise carefully. First, you should use a better title for reflecting your study. Second, you should provide more information in your abstract section. Your abstract is not very informative. Your introduction and methodology is OK. But your results and discussion needs to improve. It seems you assembled it very fast. You just indicated your results and there is not any comparison with other studies. There are studies such as Antibiotics 2021, 10(1), 87; https://doi.org/10.3390/antibiotics10010087, which you can use for improvement of your manuscript. So, I highly recommend to do a major revision on your manuscript , provide more details , specially in your results and discussion section, for reconsideration. Also, provide your statistical analysis which you used in this study.

Reviewer #2: In this manuscript, the authors investigated the morphological dynamics of microbial cell proliferation on an antimicrobial surface using E. coli on the surface of a gel incorporating AgNbO₃ particles. The manuscript is well written; however, there are some concerns as follows:

1. Regarding the last paragraph of the Introduction, "It is worth mentioning that incorporation of the right amount of AgNbO₃ nanoparticles in PMMA bone cement is expected to confer permanent antibacterial property to the cement due to its extremely limited corrosion/dissolution rate, contrary to the incorporation of conventional antibiotics such as gentamicin," this hypothesis is unclear as the manuscript does not specifically verify the "permanent antibacterial property."

2. In section 2.5, “Using imageJ software, the microscopic images were analyzed for extracting the characteristic features of three representative colonies within one spot for each condition, and the average size of the 20 representative colonies from the two spots for each condition.” It’s not comprehensible to relate “20 representative colonies” to “three representative colonies within one spot”. And the result in Table 1 showed “the average size of 10 representative colonies”.

3. Did you polish the resin specimens incorporating nanoparticles in the methods section 2.6? If so, please specify the type of polishing paper used and the grit sizes used for polishing. This information should also be included in S7 Appendix.

4. There was no icon of “20 μg/mL” in Fig.1.

5. In section 3.2, equation (1) is is not accessible.

6. In section 3.2, there was no explanation about how to relate the increase factor to division cycles.

7. In section 3.3, “However, first report our observation with the growth behavior of bacteria on the surface of gels containing sub-MIC surface concentration of antimicrobial particles.” The sentence is not comprehensible.

8. In section 3.3, “For example, at ¼ and ½ the surface MIC, i.e. gels with 5 ng/mm2 and 10 ng/mm2 nanoparticles, the lag phases are 1 and 2 h respectively (Fig 1).” The reference should be Fig. 5.

9. In Fig A-C in S6 Appendix, why the author exhibited the images after 5 h of incubation, not 6 h?

10. Regarding the statement in the results section 3.4, "The predictive capability of our findings on the gel surface MIC was tested by assessing the antimicrobial level of a composite material," evaluating the growth of planktonic bacteria in the presence of nanoparticles, as mentioned in section 2.3, is beneficial for assessing the antibacterial properties of AgNbO3 itself. However, although evaluating colony formation on an agar surface with nanoparticles, as in section 2.4, is appropriate, it is not suitable for predicting bacterial growth on materials such as PMMA. The surface characteristics of the agar medium differ from those of PMMA, and therefore, adding nanoparticles to PMMA and assessing bacterial growth on its surface cannot be considered “predictive”.

Reviewer #3: The work deals with the study of E. coli growth dynamics on a surface with AgNbO3 antimicrobials particles. In this sense, the title is too general because only E. coli but not other bacteria were evaluated.

Although the theme and objectives of the work contribute to knowledge in the area of antimicrobial materials, the manuscript have some errors or not clear sentences to correct before being considered for publication in the Journal. See attached file.

6. PLOS authors have the option to publish the peer review history of their article (what does this mean?). If published, this will include your full peer review and any attached files.

Reviewer #1: No

Reviewer #2: No

Reviewer #3: No

---

## [Author Response · Author response to Decision Letter 0]

24 Jul 2024

Dear Dr. Jang,

Thank you for the careful consideration of our manuscript. We appreciate the constructive feedback and support of you and the reviewers.

Please find enclosed our revised submission, which includes the response letter to the reviewers, a marked and unmarked copy of the revised paper, and updated figures and supporting information files.

With kind regards,

Cyrus

---

## [Decision Letter · Decision Letter 1]

6 Aug 2024

Growth dynamics of Escherichia coli cells on a surface having AgNbO3 antimicrobial particles

PONE-D-24-21073R1

Dear Dr. Talebpour,

We’re pleased to inform you that your manuscript has been judged scientifically suitable for publication and will be formally accepted for publication once it meets all outstanding technical requirements.

Kind regards,

Hae Lin Jang

Academic Editor

PLOS ONE

Reviewers' comments:

Reviewer's Responses to Questions

**Comments to the Author**

1. If the authors have adequately addressed your comments raised in a previous round of review and you feel that this manuscript is now acceptable for publication, you may indicate that here to bypass the “Comments to the Author” section, enter your conflict of interest statement in the “Confidential to Editor” section, and submit your "Accept" recommendation.

Reviewer #1: All comments have been addressed

Reviewer #2: All comments have been addressed

2. Is the manuscript technically sound, and do the data support the conclusions?

Reviewer #1: Yes

Reviewer #2: Yes

3. Has the statistical analysis been performed appropriately and rigorously? 

Reviewer #1: Yes

Reviewer #2: Yes

4. Have the authors made all data underlying the findings in their manuscript fully available?

Reviewer #1: Yes

Reviewer #2: Yes

5. Is the manuscript presented in an intelligible fashion and written in standard English?

Reviewer #1: Yes

Reviewer #2: Yes

6. Review Comments to the Author

Reviewer #1: I would like to let you know that the respected Authors considered the comments which I recommended.

Reviewer #2: (No Response)

7. PLOS authors have the option to publish the peer review history of their article (what does this mean?). If published, this will include your full peer review and any attached files.

Reviewer #1: No

Reviewer #2: No

---

## [Editor Report · Acceptance letter]

9 Aug 2024

PONE-D-24-21073R1 

PLOS ONE

Dear Dr. Talebpour, 

I'm pleased to inform you that your manuscript has been deemed suitable for publication in PLOS ONE. Congratulations! Your manuscript is now being handed over to our production team.

Kind regards, 

on behalf of

Dr. Hae Lin Jang 

Academic Editor

PLOS ONE